# Pandemic lifeworlds: A segmentation analysis of public responsiveness to official communication about Covid-19 in England

Stephen Coleman[1]*, Michael D. Slater[2], Phil Wright[3], Oliver Wright[4], Lauren Skardon[4¤a], Gillian Hayes[5¤b]

1 School of Media and Communication, University of Leeds, Columbus, Leeds, England, 2 School of Communication, The Ohio State University, Columbus, OH, United States of America, 3 Smarter Thinking, Kaitaia, New Zealand, 4 Savanta, London, England, 5 United Kingdom Health Security Agency, London, England

¤a Current address: Boxclever, Featherstone, England
¤b Current address: Independent Market Research Consultant, Liverpool, England
* s.coleman@leeds.ac.uk

**Data Availability Statement:** Data relevant to this study are available from the University of Leeds repository at https://doi.org/10.5518/1463.

## Abstract

Pandemics such as Covid-19 pose tremendous public health communication challenges in promoting protective behaviours, vaccination, and educating the public about risks. Segmenting audiences based on attitudes and behaviours is a means to increase the precision and potential effectiveness of such communication. The present study reports on such an audience segmentation effort for the population of England, sponsored by the United Kingdom Health Security Agency (UKHSA) and involving a collaboration of market research and academic experts. A cross-sectional online survey was conducted between 4 and 24 January 2022 with 5525 respondents (5178 used in our analyses) in England using market research opt-in panel. An additional 105 telephone interviews were conducted to sample persons without online or smartphone access. Respondents were quota sampled to be demographically representative. The primary analytic technique was $k$ means cluster analysis, supplemented with other techniques including multi-dimensional scaling and use of respondent - as well as sample-standardized data when necessary to address differences in response set for some groups of respondents. Identified segments were profiled against demographic, behavioural self-report, attitudinal, and communication channel variables, with differences by segment tested for statistical significance. Seven segments were identified, including distinctly different groups of persons who tended toward a high level of compliance and several that were relatively low in compliance. The segments were characterized by distinctive patterns of demographics, attitudes, behaviours, trust in information sources, and communication channels preferred. Segments were further validated by comparing the segmentation variable versus a set of demographic variables as predictors of reported protective behaviours in the past two weeks and of vaccine refusal; the demographics together had about one-quarter the effect size of the single seven-level segment variable. With respect to managerial implications, different communication strategies for each segment are suggested for each segment, illustrating advantages of rich

**Funding:** SC award from UK Health Security Agency The sponsor played a role in the study design, analysis and preparation of the manuscript, but the principal decisions were made by independent academic scholars

**Competing interests:** The authors have declared that no competing interests exist.

segmentation descriptions for understanding public health communication audiences. Strengths and weaknesses of the methods used are discussed, to help guide future efforts.

## Introduction

Faced in 2020 with Covid-19, a worldwide threat that could only be tackled through concerted public action, it was a fundamental duty of governments throughout the world to formulate and disseminate clear and practicable guidance that would reach and guide all sections of the public, regardless of background or outlook. Doing so presented a formidable communicative challenge, for the public is not a homogeneous entity and the coordination of mass collective action in the face of a common threat is bound to be complicated by the different ways in which people experience the threat and receive, interpret and act upon official guidance relating to it. There is strong empirical evidence to suggest that generic, one-size-fits-all messages about public health are less effective than ones that are sensitive to the attitudinal perspectives and experiential lifeworlds of specific social groups [1–5].

Segmentation techniques, which emerged originally as a means of differentiating commercial market demand with a view to targeting distinctive groups [6], employ cluster analysis or related methods such as latent class analysis to identify groups within a large population that exhibit patterns of responses across a complex set of variables [7]. As such, they provide an important complement to more qualitative or descriptive efforts to understand population issues in the pandemic [8, 9]. In terms of public health message targeting, breaking up a heterogeneous audience into relatively more homogeneous audiences permits message content to be geared to the life experiences, worldviews and social opportunities and constraints of specific population segments, and dissemination channels preferred by those segments.

Accordingly, there have been several audience segmentation studies in the context of the Covid-19 pandemic. Some were based on subpopulations of interest, e.g. identifying differences by race, ethnicity, and age in vaccination attitudes and behavior for Medicaid parents in Florida, USA [10], and a study of a single U.S. county based on variables available in a longitudinal study designed originally for other purposes [11]. Others were national in scope but limited in the range of variables studied: e.g. Kamenidou et al. surveyed 3359 Greek respondents and identified five segments based on a continuum of self-reported Covid protective behavior but had little additional data profiling attitudes, trust, and media use [12]. Schneider et al. used an 11-item instrument including perceptions about vaccine efficacy and politicization of Covid vaccination with a sample of 583 U.S. adults [13]. Stubenvoll focused on Covid-related misperceptions and false beliefs among a sample of 913 Austrians, providing a rich profile of acceptance and rejection of scientific (mis)information, though not of attitudes and beliefs about Covid or themselves that might inform intervention design [14]. Ihm and Lee conducted a survey of 723 South Korean adults segmenting on social resources, social support, and media use, providing provocative insights regarding well-resourced versus vulnerable populations though without accompanying psychosocial data typically included in health audience segmentation [15].

Other national surveys were more robust with respect to psychosocial determinants of health behavior that are typically recommended for incorporation into health audience segmentation analyses [7]. For example, Thaker et al. surveyed 1054 Australian adults regarding vaccine intentions and behavior with 16 items based on the Theory of Planned Behavior plus items on media, doctor, and official source trust [16, 17]. This analysis found a five-segment

solution, also representing a continuum of vaccine enthusiasm but also identifying a segment ambivalent about vaccination personally but motivated to protect the health of others. They found that trust in all channels except social media were highest for what they called Vaccine Enthusiasts (a group who were predominantly male); trust tended to decline with reduced vaccine support except for trust in social media. They called for future research to cast a wider net in terms of determinants of Covid behaviors than TPB alone. More ambitiously, Zhou, Li, and Shen drew on several theories of health determinants in addition to TPB in a segmentation study sampling 1041 Americans, again focusing on vaccine hesitancy [18]. This approach yielded relatively rich profiles, capturing some complexity in ambivalence, in different levels of risk perception, perceived vaccine efficacy, among other psychosocial variables, though lacking the data on source trust provided by Thaker et al. [16].

Our study adds to this body of research in four ways. First, it addresses a gap in existing Covid-19 audience segmentation research by combining rich psychosocial detail about audience responses—using dozens of items beyond the Theory of Planned Behaviour measures used in prior studies—with data on information source trust and use such as that used by Thaker et al, to provide the most developed audience segmentation profile concerning Covid-19 to date. Second, it is based upon a national survey of a core nationally representative sample of over 5000 English adults (with quotas set for age, gender, region, and social grade), with boosters used to achieve larger samples for ethnic minorities, providing a robust national-level profile, and one that can be usefully compared to the other efforts worldwide described above. Third, our segments are profiled against behaviours such as masking and social distancing as well as vaccination, the focus of most prior Covid segmentation research. Fourth, our study emerged from a collaboration between a national government agency that has been central to the design and implementation of health communication during the pandemic (the UK Health Security Agency, UKHSA); a leading market research survey company (Savanta) and the University of Leeds. It is rare for segmentation studies to be so closely embedded in national institutions that are responsible for critical policy actions, or to combine efforts of academics, professional market researchers, and government agencies. This combination allowed us to work with a variety of perspectives in the design and analysis of the study and may provide some lessons for such collaborations in future.

In summary, we ask the following key research questions. What are the major audience segments, drawing on relevant attitudes, motivations, and other determinants of health behavior, identifiable with respect to pandemic/Covid-19 communication in the UK? How do these segments compare with one another, with respect to key attitudes, demographics, and communication channels trusted and used?

In addition, we compare how well the segments we find predict protective public health behaviours and vaccination refusal relative to the demographic variables typically associated with such behaviours. Finally, we also discuss some methodological lessons and insights for public health managers and communicators regarding communication strategies that might be most appropriate with these various population segments.

## Methods

The University of Leeds Faculty of Arts Humanities and Culture Ethics Review Board approved this study (reference: FAHC 20–093 AHC FREC) Members of the survey panel were recruited by a third party (the market research company, Savanta) and we (the researchers) did not have access to their names or addresses. Panel members signed written consent with the third party.

## Survey data collection

The market researchers conducted 5,525 surveys online in England between 4 and 24 January 2022 and 105 surveys via telephone between 26 January and 7 March. The questions were then divided into 12 blocks of related items and for each we measured the respondent-level variability. A set of rules were devised to identify respondents who showed little or no variation in their responses (either across multiple blocks or overall). Some blocks were omitted from consideration since it was deemed that a consistent response may be reasonable (e.g. satisfaction with various aspects of ones' life). We also plotted respondent-level mean and variance for several key blocks of attitudinal statements. A key feature of the plots was the existence of a small proportion of respondents who completely agreed with (almost) every statement (seen as low variability, high mean), even when statements had contradictory meanings. These respondents were also removed from further analysis. In total 329 respondents were removed, leaving 5,178 for the segmentation analysis. The resulting 5,178 online respondents comprised a core, nationally representative sample of UK adults with quotas set for age, gender, region, and social grade, with boosters used to achieve larger samples for ethnic minorities (1,405), those in deciles 1–3 of the Index of Multiple Deprivation (1,975), and those in 20 local authorities that had seen particularly enduring levels of Covid-19 transmission (558). The additional 105 surveys conducted via landline telephone were with people who were digitally excluded (defined as never having used the internet or not having used it in the last three months), for a total of 5,283 included in the segmentation cluster analysis. 51% of the sample were female and 49% male. 83% were white British, with the remaining 17% including persons of Indian, Pakistani, Bangladeshi, and African origin as the largest ethnic minority groups.

## Development of questionnaire instrument

This project was a collaboration between the UKHSA, market researchers and academic researchers. An initial inventory of questions was developed utilizing items that had proved useful in related previous consumer research, had been used in government tracking efforts previously, or were of particular policy concern to UKHSA. These were supplemented by the academic researchers to better approximate measurement of variables of theoretical interest. Items used to create measures that shaped the segmentation analysis are discussed below; other individual items used to profile segments (e.g., vaccination status, two-week self-reported protective behaviours, media trust, and preference) are identified in the results section.

Given the eclectic development of the survey items, measures were created empirically using exploratory factor analysis. Initially the questions were divided into three thematically similar batches, to make exploratory factor analysis more manageable, and factor analysis was used to identify reliable measures. $k$-means clustering was used to generate six clusters, five of which were found to be readily interpretable, whilst the sixth appeared ambiguous and contradictory. In particular, the segment was found to agree largely with the agree/disagree Likert-type attitudinal statements (even contradictory statements)—a phenomenon known variously as courtesy or acquiescence bias [19–21]; other items, such as behavioral self-reports, did show consistency and variability across the range of the scale. Several attempts were then made to remove unreliable respondents (in some cases such patterns appeared to be due to people rushing through the questions, so we tried increasing the maximum elapsed time required to accept a respondent), but this was unable to resolve the issues with response bias posed by this segment, leading us to conclude the problem was largely due to acquiescence bias. It was of particular concern, as the segment had a disproportionately large percentage of ethnic minority respondents, and the literature suggested that cultural differences could result in differential

patterns of acquiescence bias [22–24]. I.e., in the literature, some cultural groups were disinclined to openly express outright disagreement, though the extent of agreement did vary, a pattern reflected for this group of our respondents.

To address this problem, the market research firm overseeing the segmentation analysis engaged a consultant (co-author Phil Wright) with expertise in addressing complex and difficult market research challenges, to oversee and conduct further data analysis and produce a segmentation scheme that addressed the problem of acquiescence bias among some respondents.

The first step was to standardize the data. In most cases this was to ensure that question means were zero and standard deviations were one. For two blocks of questions however (where scale usage by the respondents varied considerably and where the acquiescence bias was most evident), standardization was instead performed at a respondent level rather than question level. This adjusts for acquiescence bias by looking at variation from the respondent's own mean, thereby eliminating the problem of some respondents having a mean shifted in the positive direction due to a tendency to agree with statements. The question levels' means and standard deviations were found to be close to 0 and 1 respectively.

The standardization procedures did impact factor structures, requiring recreation of the measures. Splitting the data and iteratively creating factors yielded 33 factors. Due to the independent nature of their construction several of these were highly correlated. Since 33 factors were too many to be considered for clustering and because several themes overlapped a pragmatic decision was made to use factor analysis to combine these into just 14 'meta-factors'. These meta-factors allowed us to work with a manageable set of composite variables. The segmentation scheme was initially developed using these initial 14 meta-factors, and then we reproduced the scheme using a refined set of clearly reliable factors (described below).

## Segmentation analysis procedures

It should be noted at the outset that segmenting audiences for marketing and communication purposes is typically an iterative process that involves both judgment and empiricism. $k$-means clustering is typically used in audience segmentation, but the choice of variables used in creating such clusters, and the number of clusters selected, is based on utility and interpretability (e.g., [3]). $k$-means clustering was supplemented using multi-dimensional scaling, which can provide additional conceptual insight about underlying differences between segments.

The following is a summary of the steps taken in developing the segmentation and in determining the number of clusters to be used in our analysis. We applied $k$-means clustering to the 14 factors and then to a sub-set of these factors; however, no convincing solution emerged after several attempts. Multi-dimensional scaling was used to produce a two-dimensional plot of the different factors. Each respondent was assigned to the map (using their weighted position based upon the 14 factors) and was then assigned polar co-ordinates. Those with a radius greater than a given cut-off (varied but used to remove the most neutral of respondents) were divided into 36 sectors. Here the profiling was found to be highly predictable (a series of sine-waves each lagged by a different amount to reflect the spread of the 14 meta-factors around the plot), but unfortunately not as discriminatory as we had hoped.

Then, the above multi-dimensional scaling approach was replicated but using only two metrics (found by forcing the 14 meta-factors into two dimensions). These broadly measured (i) engagement/concern with Covid-19, and (ii) personal responsibility / ownership of ones' health. This was a little clearer but still failed to produce a workable solution. In using two dimensions, whilst easier to explain, we had moved too far away from the multi-dimensional nature of the data.

Next, *k*-means clustering was applied to the two metrics above. Here most solutions were found to be highly unstable as determined by plotting segment positions using multi-dimensional scaling for approximately 20 different clustering runs for different numbers of segments. Only one input resulted in near-identical results every time: five clusters but with the central group of about 2% of respondents removed. The central group were found to be those with average responses across most/all factors. That is, they had no strong opinions either way. Unsurprisingly, such respondents reduced the clarity of the segmentation scheme.

Using the above result as a starting point these five clusters were profiled. Four of the five were similar to the original segments created in the initial *k*-means analysis mentioned above (although in all cases a little clearer and easier to interpret). A fifth cluster was new. Of the two segments that no longer appeared in our analysis, one was the earlier identified problem segment that had been characterized by acquiescence bias, the other was an interesting segment that we wished to retain.

For each of the five new clusters we sought to identify sub-clusters using both *k*-means clustering and various sensible partitions using factors or questions. In each case nothing worth retaining was produced. The 'missing' segment was recreated by re-allocating respondents of this segment to a sixth segment (effectively creating a segment identical to the original but changing the nature of the other segments). Re-profiling suggested that–in the main–this was a positive change to the segmentation.

We then considered the central segment and agreed that this was in fact a quite reasonable segment in its own right (people without a strong or distinctive point of view on this issue). This meant we did not have to drop the central 2% of respondents. This resulted in seven segments in total.

Finally, a matrix of Euclidean distances was created between all respondents and the centers of the seven segments. By then creating a confusion matrix (comparing current and closest segments) we were able to iteratively move respondents to their nearest segment whilst examining the impact on the segment profiles as we progressed. Fortunately, when all respondents were assigned to their nearest cluster the result was sharper and clearer (because the dampening effect of the central segment was removed from the other segments). This process was effectively the same as running *k*-means clustering with just one iteration.

Figs 1 and 2 in S1 Appendix provide the figures representing the multidimensional scaling conducted to assess interpretability of the segments identified.

Having agreed on the resulting solution we then applied the segmentation to the booster sample using Quadratic Discriminant Analysis (QDA) (training accuracy = 88%). Most incorrectly allocated respondents in the training data set were assigned to adjacent segments.

An initial segmentation pass was conducted using empirically derived factors. We then set about refining factor reliability. For the core factors necessary to rebuild the segmentation (ten in total), eight had performed sufficiently well (Cronbach's alpha > 0.7), but two were well below the required threshold.

To address this and demonstrate that the resultant segmentation was based upon reliable factors the segmentation was recreated (using distance to segment centroid as described above) but using only factors that were deemed reliable. The two factors that had inadequate reliability (as tested with Cronbach's alpha) were reworked using other items until we arrived at a measure that was reasonably equivalent conceptually and was reliable. We then used the necessary ten factors and compared our results with the final segmentation using a confusion matrix [25].

Since it is nearly impossible to exactly replicate a segmentation from centroids alone (perturbing the data slightly can easily result in a similar but not exact match) an informal threshold was identified for a respondent to be moved to a new segment. This effectively prevented

respondents on the edge of a segment from making a relatively small move to the adjacent segment and thus being recorded as being assigned incorrectly. A simple bootstrap (resampling with replacement) allowed us to estimate a reasonable threshold for change. Once applied, our re-build segmentation, using entirely reliable factors, matched 98.4% with the final segmentation. We therefore concluded that the original segmentation scheme was essentially identical with one built with all factors over a .7 Cronbach alpha threshold. A list of the factor reliability scores and items comprising each factor is provided in S2 Appendix for the final refined factors. These factors are described below.

Factors were computed using factor loadings (where the direction of the items differed, this was reflected in the use of positive or negative factor loadings, which served in effect to reverse-code those items). All items comprising each factor in the EFA were included even where there was cross-loading, as the factors are used to generate the cluster analysis and are not used in associational analyses that would be problematic given such cross-loadings. The focus in understanding the clusters should be on the profiling variables not included in the factors, as described in the results below.

The first factor, *manageability of Covid-19 risks*, was constructed using 22 items (Cronbach's alpha = .864). Sample items include: "life is too short to be worrying too much about Covid-19 risks" and "based on my experience, Covid-19 is not a threat". (All items comprising this and the other factors described below are listed in S1 Appendix). The second factor, *effectiveness of protective behaviors*, was constructed using 35 items (alpha = .962). Sample items asked about effectiveness of wearing face masks, vaccines, and testing. 14 items comprised the third factor, *concern about Covid-19 risks* (alpha = .941), and included items about the extent to which Covid-19 posed a severe risk to the respondent, family members, and the UK population as a whole, as well as questions about worry overall and engaging in various activities. The fourth factor, *personal well-being* (alpha = .779), included a dozen items asking about physical, mental, and financial health. *Self-care* (alpha = .861) was the fifth factor; four items addressed efforts to maintain a healthy lifestyle, exercise, and diet. Sixth was *sociability/sensation-seeking* (alpha = .754), with six items addressing importance of socializing, enjoyment of risk-taking and novel experience, and impulsiveness. Seventh was *covid-19 self-reliance* (alpha = .71), with five items concerning personal responsibility for health and decision-making concerning Covid-19-related behaviors. *Anxiety about world* (alpha = 7.92) was eighth, with six items regarding worry about climate change, air pollution, antibiotic resistance, etc. The ninth factor was *comprehension/trust in official guidance* (alpha = 762), with three items about how easy or difficult it was to make sense of official guidance about Covid-19 and whether government and politicians had given honest and clear information. The tenth factor, *personal health anxiety* (alpha = .825), included five items concerning caution about going back to normal, concern about crowded spaces, and personal risk of Covid-19.

The refined and reliable factors were used to create the segments described and profiled in the Results section below.

## Results

### Characterizing the final segmentation solution

Given the richness and complexity of these data, we summarize our observations about each of the segments below in a narrative form, based on our review of the profiling differences. This should provide an understanding of the segment names and labels, and thus make perusal of the tables detailing differences between segments more intelligible.

We refer to our first segment as **the Trusting Compliers** (14% of the population). Members of this segment tend to follow official guidance and are able to do so without major loss or

inconvenience to their everyday lifestyles. 56% are male. One in three (33%) are in socio-economic grades A and B, which is 10% more than the population average. Their mean age is 58 (10 years older than the average for the population). 62% are in work and 54% have children in their household.

Trusting Compliers are more interested than any other segment in acquiring information about the pandemic. Most find such information easy to understand. They trust medical professionals to give them good advice and have strong trust in mainstream media, such as TV, radio and newspapers. Nine out of ten members of this segment (91%) report complying with official advice relating to the pandemic.

The second segment are **the Concerned Cooperators** (14% of the population). Members of this segment try to do what is expected of them but are not always sure about what that official advice is or whether it can be trusted. The segment gender split (51% female) reflects the national average. Over half are in socio-economic grades C1 and C2, making them quite close of the population average. Their mean age is 54. 67% are in work and 55% have children in their household.

Like the Trusting Compliers, the Concerned Cooperators are interested in acquiring information about the pandemic, but approximately two thirds of them do not find the guidance they are offered easy to understand. They tend to trust messages from the mainstream media, such as TV, radio and newspapers, and they trust medical professionals to give them good advice. 86% of the members of this segment comply with official advice relating to the pandemic.

The third segment are **the Fearful and Overwhelmed** (13% of the population). These people tend to feel scared and lost, often confused by the guidance they are offered and seeing health insecurity as one of many pressing challenges with which they must cope. At 64%, this is the most predominantly female segment. Over 1 in 3 (36%) of the people in this segment are in socio-economic grades D and E - 11% more than the population average. Their mean age is 48. 51% are in work, but only 27% have children in their household, which is 16% below the national average.

Just over half (58%) of the Fearful and Overwhelmed are interested in information about the pandemic, but over a third of them find such information difficult to understand. They have lower than average trust in guidance offered to them by medical professionals or the mainstream media. A significant proportion of this segment turn to alternative online sites for information about the pandemic. They also have a high level of trust in faith groups compared to other segments. Three in 4 (73%) people within this segment comply with official guidance relating to the pandemic.

We refer to the fourth segment as the **Informed and Responsible** (13% of the population). Members of this segment are inclined to weigh up any official advice that is given to them relating to the pandemic in accordance with their own experiences, sometimes challenging what they are being told. 60% are male. This group has a broad socio-economic distribution, with over half falling within socio-economic grades C1 and C2. Their mean age is 56. They have the highest proportion of segment members in work (72%) and having children in their household (63%).

Only a minority (37%) of Informed and Responsible are interested in information about the pandemic, but most (55%) find the official guidance they do receive easy to understand. Members of this segment tend to trust medical professionals and, to a lesser extent, mainstream media, but they question what they are told and want to be able to verify facts for themselves. Most people in this segment (72%) comply with official advice relating to the pandemic.

The fifth segment are the **Nonchalant** (15% of the population). Members of this segment tend to have no strong views about the official guidance they are offered. As pragmatists, they

are prepared to make an effort but do not always see the point of making big life changes. Just over half (55%) are female. The distribution across socio-economic grades within this group reflects the population average. Their mean age is 43. Two thirds (67%) of them are in work (10% above the population average) and 44% have children living in their household. Four in 10 (39%) members of this segment are interested in receiving information about the pandemic, but fewer than 1 in 3 (28%) find such information easy to understand. In accessing information relating to the pandemic, they tend to move between mainstream media and alternative online sites, and they trust advice from medical professionals. One in 5 of the Nonchalant group trust information from faith groups. Half of the people in this segment comply with official advice; half do not.

The sixth segment are the **Unconcerned and Uncooperative** (at 21% of the population, the largest segment). Members of this segment lead busy lives and do not want to be disturbed or held back by crisis conditions. Just over half (51%) are female. Over half are in socio-economic grades C1 and C2, making them quite close of the population average. This is the youngest of our segments, with a mean age of 36. Only half (51%) are in work and over two-thirds live in households without children.

Three-quarters of the members of this segment have no interest in acquiring information about the pandemic and over a third (35%) say that they feel overwhelmed by official advice and do not feel sure whether it is correct. They have low trust in any source of external information and are more likely to trust friendship networks than mainstream media. Only 1 in 4 members of this segment comply with official advice.

The seventh segment are the **Skeptical Resisters** (15% of the population). Members of this segment do not want to be told what to do. They outrightly resist official guidance and are happy to be considered social rebels. 53% of this group are female. Over half (55%) are in in socio-economic grades C1 and C2, with almost a third (31%) in grades D or E. Their mean age is 48. 54% are in work and 41% have children in their household.

Just over half of this segment are interested in information about the pandemic, but over 1 in 3 (36%) find it difficult to comprehend. Members of this segment have the second lowest level of trust in medical professionals. They are very skeptical about the honesty or clarity of messages from politicians or experts. One in three people within this segment trust information about the pandemic that they find on alternative online health sites and a third trust faith groups. Only 1 in 4 people in this segment comply with official advice relating to the pandemic and they have far and away the lowest rate of vaccination of any segment.

## Comparing the segments on key variables

In the tables that follow, for each variable profiled, scores were compared between each segment using a *Z* test for percentages and a *t* test for means and a .05 significance level for each test, as is standard practice in market research; the purpose is primarily to highlight differences that are worthy of attention. The focus in audience segmentation is interpreting the overall pattern of results, not (over)interpreting comparison of a few specific scores in isolation, given the number of comparisons made. Each column (corresponding to one of the segments) is assigned a letter so it is clear which segments appear to be different from each other, beginning with the largest mean or percentage, with comparison to the smaller means and percentages for that variable. Therefore, the smallest mean or percentage (or the smallest several, if they are not significantly different) will have no letter as there are no smaller cells with which to compare, and the larger cells should be examined to see which are different from the smaller cells.

Table 1 profiles the segments demographically. As one might expect given the relationship of age Covid-19 risk, segments vary quite a bit by average age. At the same time, people of the

**Table 1. Demographic profile of audience segments.**

| | Trusting Complier | Concerned Cooperator | Fearful & Overwhelmed | Informed & Responsible | Non-chalant | Unconcerned & Uncooperative | Skeptical Resister |
|---|---|---|---|---|---|---|---|
| **AGE** | | | | | | | |
| 18–24 | 2% | 5% | 8% | 5% | 14% | 25% | 7% |
| | | a | abd | a | abcdg | abcdeg | a |
| 25–34 | 7% | 11% | 18% | 9% | 24% | 27% | 19% |
| | | a | abd | | abcd | abcdg | abd |
| 35–44 | 10% | 12% | 17% | 11% | 20% | 23% | 18% |
| | | | abd | | abd | abcdg | abd |
| 45–54 | 17% | 18% | 19% | 17% | 16% | 13% | 20% |
| | f | f | f | | | | f |
| 55–64 | 21% | 21% | 19% | 20% | 12% | 6% | 16% |
| | efg | efg | ef | ef | f | | ef |
| 65–74 | 30% | 23% | 15% | 25% | 9% | 4% | 14% |
| | bcdefg | cefg | ef | cefg | f | | ef |
| 75+ | 13% | 10% | 4% | 13% | 6% | 2% | 6% |
| | cefg | cefg | f | cefg | f | | f |
| **GENDER** | | | | | | | |
| Female | 44% | 51% | 64% | 40% | 55% | 51% | 53% |
| | | ad | abdefg | | ad | ad | ad |
| Male | 56% | 49% | 36% | 60% | 45% | 49% | 47% |
| | bcefg | c | | bcefg | c | c | c |
| **RACE** | | | | | | | |
| NET: White | 90% | 88% | 88% | 91% | 81% | 81% | 92% |
| | ef | ef | ef | bcef | | | bcef |
| NET: Black | 2% | 3% | 2% | 2% | 4% | 5% | 3% |
| | | | | | cd | abcdg | |
| NET: Asian | 6% | 7% | 7% | 5% | 12% | 9% | 3% |
| | g | g | g | g | abcdg | abdg | |
| NET: Other | *% | 1% | 1% | *% | 1% | 2% | *% |
| | | | ag | | g | adg | |
| **SEG** | | | | | | | |
| A, B | 33% | 22% | 17% | 28% | 23% | 20% | 17% |
| | bcdefg | cg | | bcefg | cg | | |
| C1, C2 | 49% | 54% | 47% | 51% | 54% | 55% | 53% |
| | | c | | | ac | ac | |
| D, E | 18% | 24% | 36% | 20% | 23% | 25% | 31% |
| | | a | abdef | | a | ad | abdef |

Note: see text for an explanation of the subscripts indicating significant differences between segments for this and all tables following.

same age can be in segments that are dramatically different with respects to pandemic-related attitudes and behaviors, as is evident below. Definitions of socioeconomic segments are described in Methods.

Table 2 demonstrates the sometimes quite dramatic differences between segments with respect to vaccine compliance and cooperation regarding masking and social distancing in the two weeks prior to responding to the survey.

**Table 2. Vaccination status and past two-week protective behavior by segment.**

| | Trusting Complier | Concerned Cooperator | Fearful & Overwhelmed | Informed & Responsible | Non-chalant | Unconcerned & Uncooperative | Skeptical Resister |
|---|---|---|---|---|---|---|---|
| **Vaccination Status (self-report)** | | | | | | | |
| I have had both initial vaccinations and a booster (three total) | 88% | 80% | 72% | 83% | 59% | 39% | 46% |
| | bcdefg | cefg | efg | cefg | fg | | f |
| I have had both initial vaccinations but not a booster | 9% | 13% | 16% | 11% | 21% | 22% | 19% |
| | | a | ad | | abcd | abcd | abd |
| I haven't had a vaccination and don't intend to | *% | 1% | 3% | 2% | 2% | 8% | 24% |
| | | a | abd | a | ab | abcde | abcdef |
| **Past two week masking, hand washing, and social distancing** | | | | | | | |
| Worn a face mask (over nose and mouth) | 96% | 92% | 88% | 92% | 76% | 53% | 69% |
| | bcdefg | cefg | efg | cefg | fg | | f |
| Washed your hands for 20 seconds or more | 90% | 87% | 81% | 80% | 62% | 46% | 50% |
| | cdefg | cdefg | efg | efg | fg | | |
| Socially distanced when in the presence of others | 86% | 77% | 73% | 62% | 55% | 35% | 28% |
| | bcdefg | defg | defg | efg | fg | g | |

In Table 3, we show segment differences with respect to some key attitudes characterizing each segment, which provide richer insight into what perspectives must be addressed when communicating with members of the segment.

Differences by segment in media channels they rely on for pandemic information are summarized in Table 4, and differences in trust of information sources of Covid-19 guidance are summarized in Table 5. Understanding these differences is crucial in determining how best to reach each segment with public health information using the information channels they are most likely to actually use for such information and which channels they are most likely to trust.

Patterns of information channel trust also provide insights into the degree of alienation or engagement with mainstream social influences such as physicians and national media sources.

## Validation: Comparing segments to demographic variables as predictors of protective behavior and vaccination refusal

One way to demonstrate the power of this segmentation analysis is using predictive validity. Public health researchers tend to look carefully at demographic influences and demographic differences, for obvious reasons, when characterizing population health data. At least some demographic variables should be strongly predictive of key Covid-19 outcome measures, such as adoption of protective behaviours and vaccine refusal. After all, Covid-19 risks are closely associated with age [26]; education may well be associated with understanding the public health value of vaccination and protective behaviours [27]; and income may influence other risk factors such as the necessity of working in high-exposure settings [28]. We can compare the segmentation scheme, which is simply one seven-level categorical variable, with these demographic variables, along with race/ethnicity and gender, with respect to prediction of protective behaviours and vaccine refusal (we note that this is a post-hoc analysis conducted in response to a reviewer query).

We created a summative measure of self-reported protective behaviours engaged in over the past two weeks prior to responding to the survey. These included masking, social distancing, avoiding crowded indoor spaces, handwashing for at least 20 seconds, and opening a

**Table 3. Selected attitudes by segment.**

| | Trusting Complier | Concerned Cooperator | Fearful & Overwhelmed | Informed & Responsible | Non-chalant | Unconcerned & Uncooperative | Skeptical Resister |
|---|---|---|---|---|---|---|---|
| **I don't want responsibility I'd rather be told what to do** | | | | | | | |
| Agree | 4% | 12% | 17% | 9% | 13% | 23% | 8% |
| | | a | abdeg | a | adg | abcdeg | a |
| Neutral | 44% | 52% | 50% | 49% | 56% | 60% | 40% |
| | | ag | ag | g | acdg | abcdeg | |
| Disagree | 52% | 36% | 33% | 42% | 31% | 17% | 51% |
| | bcdef | ef | f | bcef | f | | bcdef |
| **The future is too uncertain for a person to make serious plans** | | | | | | | |
| Agree | 14% | 27% | 48% | 13% | 26% | 32% | 24% |
| | | ad | abdefg | | ad | abdeg | ad |
| Neutral | 57% | 59% | 47% | 64% | 64% | 61% | 56% |
| | c | c | | abcg | abcg | cg | c |
| Disagree | | 14% | 5% | 23% | 11% | 7% | 21% |
| | bcdefg | cef | | bcef | cf | | bcef |
| **I feel pretty powerless when it comes to determining the future of my country** | | | | | | | |
| Agree | 27% | 45% | 66% | 27% | 36% | 35% | 45% |
| | | adef | abdefg | | ad | ad | adef |
| Neutral | 60% | 48% | 29% | 62% | 56% | 57% | 41% |
| | bcg | cg | | bcefg | bcg | bcg | c |
| Disagree | 13% | 7% | 5% | 10% | 8% | 7% | 14% |
| | bcef | | | bcf | c | | bcef |
| **I prefer to be thought of as an individual than as a member of a community** | | | | | | | |
| Agree | 30% | 35% | 36% | 33% | 35% | 32% | 47% |
| | | a | a | | a | | abcdef |
| Neutral | 60% | 55% | 57% | 61% | 58% | 61% | 47% |
| | bg | g | g | bg | g | bg | |
| Disagree | 10% | 10% | 7% | 6% | 6% | 7% | 6% |
| | cdefg | cdefg | | | | | |
| **I believe that whether or not I get Covid-19 is determined by chance and not by any action I may take or don't take** | | | | | | | |
| Agree | 83% | 68% | 57% | 60% | 43% | 33% | 40% |
| | bcdefg | cdefg | efg | efg | f | | f |
| Disagree | 17% | 32% | 43% | 40% | 57% | 67% | 60% |
| | | a | ab | ab | abcd | abcdeg | abcd |
| **We need a strong government to tell us what to do during the Covid-19 crisis** | | | | | | | |
| Agree | 75% | 77% | 66% | 56% | 50% | 31% | 33% |
| | cdefg | cdefg | defg | efg | fg | | |
| Neutral | 24% | 23% | 30% | 40% | 46% | 63% | 43% |
| | | | ab | abc | abcd | abcdeg | abc |
| Disagree | 1% | *% | 4% | 4% | 4% | 7% | 24% |
| | b | | ab | ab | ab | abcde | abcdef |

**Table 4. Covid-19 information sources by segment.**

| Where do you get your main information and updates about the Covid-19 situation? | Trusting Complier | Concerned Cooperator | Fearful & Overwhelmed | Informed & Responsible | Non-chalant | Unconcerned & Uncooperative | Skeptical Resister |
|---|---|---|---|---|---|---|---|
| Television news channels | 81% | 72% | 61% | 66% | 52% | 29% | 42% |
| | bcdefg | cdefg | efg | efg | fg | | f |
| Online news websites or apps | 49% | 45% | 41% | 35% | 33% | 23% | 30% |
| | cdefg | defg | defg | f | f | | f |
| NHS websites | 46% | 44% | 42% | 28% | 36% | 29% | 15% |
| | defg | defg | defg | g | dfg | g | |
| Government websites (local or national) | 48% | 43% | 31% | 28% | 28% | 20% | 15% |
| | cdefg | cdefg | fg | fg | g | g | |
| Friends and family | 25% | 27% | 33% | 18% | 33% | 28% | 30% |
| | d | d | abdf | | abdf | d | ad |
| Social media websites or apps | 15% | 20% | 25% | 13% | 23% | 26% | 24% |
| | | ad | abd | | ad | abd | ad |
| Print newspapers | 26% | 23% | 15% | 25% | 16% | 13% | 12% |
| | cefg | cefg | | cefg | | | |
| Work colleagues | 7% | 8% | 11% | 7% | 14% | 16% | 8% |
| | | | ad | | abdg | abcdg | |
| Print or online magazines | 8% | 8% | 9% | 5% | 10% | 11% | 4% |
| | dg | dg | dg | | dg | dg | |
| Foreign/international news outlets | 8% | 5% | 5% | 2% | 7% | 8% | 4% |
| | bcdg | d | d | | dg | bcdg | |
| Faith / community groups | 1% | 2% | 1% | 1% | 6% | 8% | 1% |
| | | | | | abcdg | abcdg | |
| Other | 3% | 1% | 2% | 3% | 2% | 1% | 3% |
| | bf | | f | bf | f | | bf |
| I don't get any information or updates about the Covid-19 situation | 0% | *% | 2% | 1% | 1% | 5% | 13% |
| | | | ab | ab | ab | abcde | abcdef |

window for ventilation when indoors with others. Vaccine refusal was based on responding affirmatively to the statement "I haven't had a vaccination and don't intend to" when offered a series of statements characterizing respondents' Covid-19 vaccination status. (At the time of

**Table 5. Trust in information sources re Covid-19 guidance by segment.**

| | Trusting Complier | Concerned Cooperator | Fearful & Overwhelmed | Informed & Responsible | Non-chalant | Unconcerned & Uncooperative | Skeptical Resister |
|---|---|---|---|---|---|---|---|
| Trust information from your GP and other medical providers | 82% | 73% | 57% | 63% | 52% | 31% | 28% |
| | bcdefg | cdefg | fg | cefg | fg | | |
| Trust information from mainstream media such as television newspapers and radio | 57% | 46% | 28% | 38% | 33% | 24% | 11% |
| | bedefg | cdefg | g | cfg | fg | g | |
| Trust information from news websites that provide an alternative to mainstream/traditional news sources | 25% | 24% | 17% | 20% | 27% | 23% | 14% |
| | Cdg | cdg | | g | edfg | eg | |
| Politicians and government officials have given honest and clear information in their official guidance about Covid-19 | 35% | 21% | 11% | 30% | 21% | 21% | 6% |
| | beefg | eg | g | bcefg | cg | cg | |

the survey booster roll-out was not complete, so that outcome would have been confounded with availability of the booster to various parts of the population at that time.) These outcome measures, of course, were not included in the items used to create the segmentation scheme. We used GLM regressions to analyze prediction of the protective behaviours (Unianova) and generalized linear model logistic regression to examine prediction of vaccine refusal (SPSS version 29).

For our summative measure of self-reported protective behaviors, the segmentation variable had a partial eta-squared of .145, which is generally considered a strong effect size [29]. Age had a partial eta-squared of .015, considered a weak effect size. All the other demographic variables each had a nominal partial eta-squared under .01. In total, the demographic variables had a partial eta-squared of .035, less than one-quarter the explanatory power of the single segmentation variable (see Table 6).

Logistic regression, needed to test prediction of vaccine refusal, does not lend itself to effect size estimation, but comparison of Wald chi-square statistics can be informative. Gender, education, and race/ethnicity were not significantly predictive of vaccine refusal. The Wald chi-square (with one *df*) was 19.5 for age and 34.5 for income. For the segmentation variable (with 6 *df*) the Wald chi-square was 459.2. As can be seen from Table 2, this was clearly due to strongly disproportionate refusal among the Skeptical Resisters and to a lesser extent the Unconcerned and Uncooperative segments.

## Discussion

Segmentation analyses such as this are intended typically to guide communication efforts. A key value of this segmentation is that public health communicators can gear message design and channel strategies regarding the pandemic to specific groups within the population.

### New insights from this research regarding Covid audience segmentation

As described earlier, prior research has either focused on specialized subpopulations [10, 11], used limited items to create the segmentation that did not permit psychological profiling [12–15], or, in the most complete available segmentation studies, used a single theoretical framework along with trust items [16], or used a richer set of psychosocial items but lacked media and source trust data [18]. These studies all focused on vaccination uptake with little or no information available on other compliance behaviors.

As summarized above, the present study combines the rich psychosocial data of the Zhou et al. study [18], and even more extensive information on media and source use and trust as found in Thaker et al. [16], and richer data on compliance beyond vaccination alone. In so doing, we believe we generated segment insights unavailable from previous research. For example, we identified four very distinct groups of cooperators - those who were well-resourced, seeking information, trusting reliable sources; those who were less well-resourced and struggling more with understanding guidance; and those who were fearful, confused, but willing to comply given their fears, and a fourth group that were also well-resourced, but were more skeptical, less trusting, though generally compliant. Among those who were less compliant, one group (with about half generally compliant with recommendations), seeming generally unengaged with the issue; a second and quite large segment was predominantly young people less at risk, more likely to trust friends and social media relative to traditional sources than other segments, with about 25% generally compliant; and a final segment distinguished by lack of trust in health authorities, government, dislike of authority - and almost one-quarter of them have not been and do not intend to be vaccinated, compared to 8% for the youthful uncooperative segment and under 5% for each of the other segments. Interestingly, other

**Table 6. Segments and demographic variables predicting Covid-19 protective behaviors.**

**Dependent Variable: Past 2 week summative measure of protective behaviors.**

| Source | Type III Sum of Squares | df | Mean Square | F | Sig. | Partial Eta Squared |
|---|---|---|---|---|---|---|
| Corrected Model | 2881.152[a] | 14 | 205.797 | 96.348 | 0.000 | 0.218 |
| Intercept | 1462.938 | 1 | 1462.938 | 684.902 | 0.000 | 0.124 |
| Segments | 1751.359 | 6 | 291.893 | 136.655 | 0.000 | 0.145 |
| Gender | 94.174 | 1 | 94.174 | 44.089 | 0.000 | 0.009 |
| Dummy_Ethnicity | 5.820 | 4 | 1.455 | 0.681 | 0.605 | 0.001 |
| EducationSelf | 27.807 | 1 | 27.807 | 13.018 | 0.000 | 0.003 |
| Income | 72.367 | 1 | 72.367 | 33.880 | 0.000 | 0.007 |
| Age | 153.335 | 1 | 153.335 | 71.787 | 0.000 | 0.015 |
| Error | 10340.288 | 4841 | 2.136 | | | |
| Total | 68737.000 | 4856 | | | | |
| Corrected Total | 13221.440 | 4855 | | | | |

a. R Squared = .218 (Adjusted R Squared = .216)

forms of compliance are quite similar for the last two segments - the last segment is uniquely resistant to vaccination in particular.

## Implications for public health managers and communicators regarding reaching these segments

Our summary of the segments, above, lends itself to some preliminary suggestions regarding public health communication strategy. For example, as the Trusting Compliers are already responsive to existing messages and communication strategies, the most effective way to target them probably is to persist in providing clear, consistent messaging through mainstream media channels. While clearly a cooperative group, the Concerned Cooperators differ from the Trusting Compliers in finding it challenging to understand official guidance. For them there is a need to simplify and pre-test messaging and maximize its consistency. Like the first two segments, the Fearful/Overwhelmed tend to comply with official guidance, but they differ from them significantly in their levels of fear and anxiety. Interpersonal and community channels, such as worksites, unions and churches could have potential for reaching and reassuring this segment by providing interpersonal support while conveying the message. For the Informed and Responsible, an emphasis upon taking protective action as an act of civic solidarity may prove to be a good fit, as would links to in-depth online information that is open to scrutiny.

The Nonchalant is an interesting segment, given its split on adherence to official guidance. Clarity and consistency of messaging is particularly important for them, with an emphasis upon relevance to their everyday life experiences. For example, members of this segment are more likely than others to have older parents, so emphasis upon the impact of behavioral choices upon others might be helpful.

Being the most resistant to official guidance, the final two segments (the Unconcerned and Unsupportive and the Skeptical Resisters) are most in need of messages that are clear, consistent, and simple. For the Unconcerned and Unsupportive, supplementing appeals via the usual channels with information dissemination within organizations in which they live and work would fit with the trust that they place in interpersonal relationships. The Skepticals might seem to be inherently averse to any kind of official guidance, but some of this at least might be due to its unintelligibility, and that could be addressed through a greater focus upon

clarity, simplicity and consistency. Lack of trust in virtually any communication channel is the biggest obstacle to reaching the Skeptical Resister. With the Unconcerned and Unsupportive, and Fearful and Overwhelmed, direct communication in interpersonal networks and organizations in which they live and work are likely to be most effective. Likewise, if possible, finding sources such as prominent individuals who the Unconcerned and Unsupportive find relatively trustworthy, and others for the Skeptical Resisters, who are willing publicly to make a case for at least some public health recommendations may prove helpful.

Of course, the above recommendations can only be broadly strategic. We possess a wealth of data from our study that could enrich communication planning and guide the development of specific messages for various segments (available at the following link: to be inserted). In addition, there is a need for in-depth qualitative investigation of how members of these segments experience and think about the world. Particularly for public health communication purposes, it is readily possible to draft messages and channel strategies for each segment, and then pretest the messages and get feedback on the channel strategy from representatives of each segment. This is normally done by conducting discriminant analyses to identify a much smaller number of questions (typically between 20 and 30 items) that can correctly identify segment members with reasonable accuracy. This short instrument can be used to identify and recruit segment members for focus groups, interviews, and other qualitative and pretesting research. The development of this reduced set of items to identify segment members for qualitative and pretest research is currently under way.

In addition to the specificity of the pandemic as a critical moment of health insecurity, we think that this segmentation can be read as a study of how a particular population (the English) form attitudes and make choices in response to a crisis that calls for their civic engagement. Unlike market segmentation analyses, which seek to split populations into sub-groups that can be ranked in terms of their value to sellers of goods and services, our aim was to find ways of appealing to peoples' commonality as citizens constituting a mutually interdependent public. The Covid-19 pandemic was a classic case of all citizens being threatened by a common risk, albeit unevenly distributed in its effects, and it was only by finding ways of persuading them to act as a public, albeit differentiated by experiential and attitudinal differences, that security could be realized. In this sense, the pandemic could be seen not only as an immediate challenge for health communicators, but as an historical template for the challenges facing collective civic action in an age of public fragmentation. Methodologically, there are also important lessons here for public health officials and health researchers. A survey with a culturally diverse national or regional population may well run into the problems of courtesy or acquiescence bias for some subpopulations, as we did, which cause difficulties for person-centered analytic procedures. Researchers should be vigilant about reviewing their data for such patterns. One option, of course, is to drop such respondents from the analysis—but if so, many members of the social group involved will no longer be represented, and often, as in our case, this may be a minority population of important public health interest. An alternative approach used here is, for the sets of items reflecting such bias, to standardize those variables based on the respondent's own mean, as described in the Methods section here.

## Limitations

Finally, we acknowledge the limitations of this study. Segmentation is always a function of the items in a data set. The data we gathered for this segmentation reflected a wide variety of items originating from the government agency that initiated it, the academics who helped to design it, and the market research company that implemented it. This convergence of perspectives was a strength insofar as it attached the study to real-world priorities, but we recognize that a

cost of this eclectic approach was that we were not primarily focused upon the use of validated academic measures and constructs in the instrument design. This has made it harder to relate our findings to the theoretical literature but may have been an advantage in that we were not bound by previous thinking in working towards creative solutions in this unique public health challenge.

This study is also a snapshot of population attitudes and behaviors at a given time point in history. While we suspect the segments would likely reproduce at many points during the pandemic, specific levels of responses to attitude and behavior questions would have been likely to vary.

We also acknowledge, as discussed above, that our analytic approach was made more challenging by the existence of a segment with acquiescence bias. The initial *k* means segmentation was relatively standard, carefully constructed on the basis of reliable factors. The subsequent one involved a state-of-the-art commercial segmentation effort which made greater use of interpretation and judgment in refining the segmentation scheme, as well as using respondent standardization where needed, to address the acquiescence bias. This resulted in a more refined, but less orthodox segmentation model, and provides a methodological approach that may prove valuable for other public health contexts in which culturally diverse populations prove to have distinctive response sets that must be addressed in a cluster analysis.

Finally, there are a great many variables profiled in these analyses and a great many comparisons made between segments. While one can have reasonable confidence in overall patterns of results, given the risk of some spurious results over so many comparisons, we recommend caution in interpreting any one comparison in isolation.

We conclude by noting that there is no one 'right' segmentation model that encapsulates the complete story. However, our combination of extensive psychosocial, behavioral, source and media trust, and compliance data we believe provides a range of insights unavailable from previous Covid segmentation studies and that may provide guidance for segmentation studies for future public health emergencies.

## Supporting information

**S1 Appendix. Multidimensional scaling for preliminary assessment of segment interpretability.**
(ZIP)

**S2 Appendix. Factors used to create segmentation and items comprising them.**
(ODS)

## Author Contributions

**Conceptualization:** Stephen Coleman, Michael D. Slater, Gillian Hayes.

**Data curation:** Oliver Wright, Lauren Skardon.

**Formal analysis:** Stephen Coleman, Michael D. Slater, Phil Wright, Lauren Skardon.

**Funding acquisition:** Stephen Coleman.

**Investigation:** Stephen Coleman, Michael D. Slater, Gillian Hayes.

**Methodology:** Michael D. Slater, Phil Wright.

**Project administration:** Gillian Hayes.

**Resources:** Gillian Hayes.

**Supervision:** Stephen Coleman.

**Writing – original draft:** Stephen Coleman, Michael D. Slater.

**Writing – review & editing:** Stephen Coleman, Michael D. Slater.

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
