## [Decision Letter · Decision Letter 0]

24 Oct 2023

PONE-D-23-27498PANDEMIC LIFEWORLDS A SEGMENTATION ANALYSIS OF PUBLIC RESPONSIVENESS TO OFFICIAL COMMUNICATION ABOUT COVID-19 IN ENGLANDPLOS ONE

Dear Dr. Coleman,

Thank you for submitting your manuscript to PLOS ONE. After careful consideration, we feel that it has merit but does not fully meet PLOS ONE’s publication criteria as it currently stands. Therefore, we invite you to submit a revised version of the manuscript that addresses the points raised during the review process.

The study is interesting and  provides valuable insights into the diverse segments of the population and their responses to public health communication strategies. It's clear that the authors have carefully considered the implications of their findings for public health communication.  This approach acknowledges the individual characteristics and needs of different groups, which is essential in effective public health communication. Overall, the study offers a practical and well-informed guide for public health communication strategy development, and it highlights the importance of considering the unique characteristics and concerns of each segment of the population.

Furthermore, it is necessary to carefully review and address the reviewer comments.

We look forward to receiving your revised manuscript.

Kind regards,

Prabhat Mittal, Ph.D.

Academic Editor

PLOS ONE

Journal Requirements:

2. Please expand the acronym “UKHSA” (as indicated in your financial disclosure) so that it states the name of your funders in full.

5. Please amend your list of authors on the manuscript to ensure that each author is linked to an affiliation. Authors’ affiliations should reflect the institution where the work was done (if authors moved subsequently, you can also list the new affiliation stating “current affiliation:….” as necessary).

Reviewers' comments:

Reviewer's Responses to Questions

**Comments to the Author**

1. Is the manuscript technically sound, and do the data support the conclusions?

Reviewer #1: Yes

Reviewer #2: Yes

2. Has the statistical analysis been performed appropriately and rigorously? 

Reviewer #1: Yes

Reviewer #2: Yes

3. Have the authors made all data underlying the findings in their manuscript fully available?

Reviewer #1: Yes

Reviewer #2: Yes

4. Is the manuscript presented in an intelligible fashion and written in standard English?

Reviewer #1: Yes

Reviewer #2: Yes

5. Review Comments to the Author

Reviewer #1: Review report of the article “Pandemic life worlds a segmentation analysis of public responsiveness to official communication about covid-19 in England”

By

Stephen Coleman, Michael D Slater, Phil Wright, Oliver Wright, Lauren Skardon, Gillian Hayes

This article presents the findings of a comprehensive audience segmentation study conducted in England, sponsored by the United Kingdom Health Security Agency (UKHSA) and executed through a collaborative effort between market research professionals and academic experts. The study employs various methods and reports results that are compared with analogous audience segmentation endeavours worldwide in the context of the COVID-19 pandemic. The article briefly outlines multiple communication strategies tailored to distinct audience segments and underscores the benefits of detailed segmentation descriptions in enhancing our understanding of public health communication audiences.

This study offers valuable insights for future research in this domain and, therefore, I recommend its publication in the PLOS ONE journal, contingent upon the authors addressing the following comments:

1. Abstract Revision: It is advisable for the authors to omit the headings such as "Background," "Methods," "Results," and "Conclusion" in the Abstract section, as these aspects are already comprehensively covered in subsequent sections of the article.

2. Citation Enhancement: To provide readers with access to the latest developments in this field and to complete the reference section, I suggest that the authors include the following references in their manuscript:

a) These revisions will enhance the clarity and conciseness of the article, making it a valuable addition to the PLOS ONE journal.

b) Tiwari, S. (2022). Impact of COVID -19 Era on Supply Chain Management and Logistics of Flipkart Company. VEETHIKA-An International Interdisciplinary Research Journal, 8(4), 25-27.

c) Gupta, S., & Tiwari, S. (2023). New Technological Advancements and Its Impact on Healthcare System. VEETHIKA-An International Interdisciplinary Research Journal, 9(1), 27-32.

d) Suresh K.P., Kumar M., Shinli V. 2022. Tribal Healthcare System in Kerala during the Pandemic, Dera Natung Government College Research Journal, 7, 83-93. DOI: https://doi.org/10.56405/dngcrj.2022.07.01.09

e) Siriwardhanaa W.S.N, Rathnayakab R.M.S.S. 2022. Awareness of Counseling Psychology and the Significance of Counseling Service for the Graduate Studies, Dera Natung Government College Research Journal, 7, 70-75. DOI: https://doi.org/10.56405/dngcrj.2022.07.01.07

f) Bosumatary S. 2019. Social Inequality and Health: A Study of Tribes in Assam, Dera Natung Government College Research Journal, 4, 7-14. DOI: https://doi.org/10.56405/dngcrj.2019.04.01.02

These revisions will enhance the clarity and conciseness of the article, making it a valuable addition to the PLOS ONE journal.

Reviewer #2: 1. No Clarity is found regarding sample size i.e. 5525.

2. The research Objectives are not clear and deviating from the aim of the research.

3. Mention the cluster's name rather than basis of segmentation .

4. Literature review is weak and no identification of Gap Analysis found .

5. How number of Clusters formed (i.e. 7),were confirmed?

6. How one cluster is different from other?, no clarity found .

7. Demographic profile of one segment is different from other should have confirmed by applying ANOVA/Chi Square, t test.

8. There is no citation included in discussion part.

9. The manuscript is not structured as per the journal guideline.

10. What is the Managerial implication of the study?

6. PLOS authors have the option to publish the peer review history of their article (what does this mean?). If published, this will include your full peer review and any attached files.

Reviewer #1: No

Reviewer #2: No

---

## [Author Response · Author response to Decision Letter 0]

18 Nov 2023

REVIEWER 1: 

1. Abstract Revision: It is advisable for the authors to omit the headings such as "Background," "Methods," "Results," and "Conclusion" in the Abstract section, as these aspects are already comprehensively covered in subsequent sections of the article.

We have edited the abstract as recommended by the reviewer, along with a few edits to further tighten it and add reference to an addition to the ms per Reviewer 2.

2. Citation Enhancement: To provide readers with access to the latest developments in this field and to complete the reference section, I suggest that the authors include the following references in their manuscript:

a) These revisions will enhance the clarity and conciseness of the article, making it a valuable addition to the PLOS ONE journal.

b) Tiwari, S. (2022). Impact of COVID -19 Era on Supply Chain Management and Logistics of Flipkart Company. VEETHIKA-An International Interdisciplinary Research Journal, 8(4), 25-27.

c) Gupta, S., & Tiwari, S. (2023). New Technological Advancements and Its Impact on Healthcare System. VEETHIKA-An International Interdisciplinary Research Journal, 9(1), 27-32.

d) Suresh K.P., Kumar M., Shinli V. 2022. Tribal Healthcare System in Kerala during the Pandemic, Dera Natung Government College Research Journal, 7, 83-93. DOI: https://doi.org/10.56405/dngcrj.2022.07.01.09

e) Siriwardhanaa W.S.N, Rathnayakab R.M.S.S. 2022. Awareness of Counseling Psychology and the Significance of Counseling Service for the Graduate Studies, Dera Natung Government College Research Journal, 7, 70-75. DOI: https://doi.org/10.56405/dngcrj.2022.07.01.07

f) Bosumatary S. 2019. Social Inequality and Health: A Study of Tribes in Assam, Dera Natung Government College Research Journal, 4, 7-14. DOI: https://doi.org/10.56405/dngcrj.2019.04.01.02

These revisions will enhance the clarity and conciseness of the article, making it a valuable addition to the PLOS ONE journal.

Thank you for these interesting citations, which we have now perused. They provide valuable insights on various aspects of healthcare systems, technology, etc. One in particular, the Suresh et al (2022) article, addresses population pandemic issues and is now cited in our ms. The others, as they do not address issues regarding population pandemic communication, or audience segmentation methodology, would risk putting us out of compliance with scholarly expectations regarding keeping citations well-aligned to the manuscript focus.

Reviewer 2: 

1. No Clarity is found regarding sample size i.e. 5525.

Please note the following under Methods, Survey data collection:

“The market researchers conducted 5,525 surveys online in England between 4 and 24 January 2022 and 105 surveys via telephone between 26 January and 7 March. The questions were then divided into 12 blocks of related items and for each we measured the respondent-level variability. A set of rules were devised to identify respondents who showed little or no variation in their responses (either across multiple blocks or overall). Some blocks were omitted from consideration since it was deemed that a consistent response may be reasonable (e.g. satisfaction with various aspects of ones’ life). We also plotted respondent-level mean and variance for several key blocks of attitudinal statements. A key feature of the plots was the existence of a small proportion of respondents who completely agreed with (almost) every statement (seen as low variability, high mean), even when statements had contradictory meanings. These respondents were also removed from further analysis. In total 329 respondents were removed, leaving 5,178 for the segmentation analysis. The resulting 5,178 online respondents comprised a core, nationally representative sample of UK adults (with quotas set for age, gender, region, and social grade, with boosters used to achieve larger samples for ethnic minorities (1,405), those in deciles 1-3 of the Index of Multiple Deprivation (1,975), and those in 20 local authorities that had seen particularly enduring levels of Covid-19 transmission (558)). The additional 105 surveys conducted via landline telephone were with people who were digitally excluded (defined as never having used the internet or not having used it in the last three months). 51% of the sample were female and 49% male. 83% were white British, with the remaining 17% including persons of Indian, Pakistani, Bangladeshi and African origin as the largest ethnic minority groups.”

2. The research Objectives are not clear and deviating from the aim of the research.

Thank you for pointing out the lack of clarity in the way we expressed the research objectives. In rereading them, we see that we’ve provided too little indication of what we will report in results and provided too much emphasis on some points we would make in discussion. We’ve reviewed these carefully and have revised that section thoroughly (see text).

3. Mention the cluster's name rather than basis of segmentation .

We are not sure of what the reviewer is referring to here. Perhaps we have addressed this issue in our reorganization of Results to address the lack of clarity the reviewer pointed out in point #6.

4. Literature review is weak and no identification of Gap Analysis found .

Per the reviewer’s recommendation, we have clarified the gap in the literature that we are addressing; we agree with the reviewer’s assessment that we had not made that point sufficiently clear in the prior version. We have added:

“Our study adds to this body of research in four ways. First, it addresses a gap in existing Covid-19 audience segmentation research by combining rich psychosocial detail about audience responses—using dozens of items beyond the Theory of Planned Behaviour measures used in prior studies—with data on information source trust and use such as that used by Thaker et al (2023, to provide the most developed audience segmentation profile concerning Covid-19 to date…”

We continue to note the other contributions: the first major characterization of the English population with respect to pandemic communication; profiling against protective behaviours as well as vaccination; and the experience of a collaboration between government, commercial market research, and academics. Regarding review scope: we have at this point incorporated in our review Covid-19 audience segmentation studies from various nations around the world, spanning four continents. Indeed we know of no other article in which these are all identified and their relative scope noted. If there are any quantitative audience segmentation studies we missed, we would love to know about them! We believe we’ve mentioned all the ones readily identifiable through scholarly search engines. It is not a large research base given the importance of the topic, which is one reason we believe our contribution is a useful one. 

5. How number of Clusters formed (i.e. 7),were confirmed?

Good question. The answer is complex. We have revised our introduction to our section on Segmentation Analysis Procedures to make it clearer to the reader how the number of clusters is selected. Again, we emphasize that in k-means analyses, it is not a matter of a “correct” solution, but finding the most useful and interpretable solution. The opening of this section now reads:

“It should be noted at the outset that segmenting audiences for marketing and communication purposes is typically an iterative process that involves both judgment and empiricism. k-means clustering is typically used in audience segmentation, but the choice of variables used in creating such clusters, and the number of clusters selected, is based on utility and interpretability (e.g., see Maibach7). k-means clustering was supplemented using multi-dimensional scaling, which can provide additional conceptual insight about underlying differences between segments.

The following is a summary of the steps taken in developing the segmentation and in determining the number of clusters to be used in our analysis.”

This following section is lengthy and technical, but I’ve excerpted here the paragraphs towards the end where he explains how we arrived at the final number of clusters (I would note that the team member responsible for this is considered one of the world’s leading market research experts on segmentation, so I think it is instructive to read and learn how he goes about making these determinations—the rest of us learned quite a bit from his approach, and I suspect other journal readers with a serious interest in audience segmentation will do so as well):

“Using the above result as a starting point these five clusters were profiled. Four of the five were similar to the original segments created in the initial k-means analysis mentioned above (although in all cases a little clearer and easier to interpret). A fifth cluster was new. Of the two segments that no longer appeared in our analysis, one was the earlier identified problem segment that had been characterized by acquiescence bias, the other was an interesting segment that we wished to retain.

For each of the five new clusters we sought to identify sub-clusters using both k-means clustering and various sensible partitions using factors or questions. In each case nothing worth retaining was produced. The ‘missing’ segment was recreated by re-allocating respondents of this segment to a sixth segment (effectively creating a segment identical to the original but changing the nature of the other segments). Re-profiling suggested that – in the main – this was a positive change to the segmentation.

We then considered the central segment and agreed that this was in fact a quite reasonable segment in its own right (people without a strong or distinctive point of view on this issue). This meant we did not have to drop the central 2% of respondents. This resulted in seven segments in total.”

6. How one cluster is different from other?, no clarity found .

We have reread the ms to try to understand the problem and we think we now see the clarity issue to which the reviewer refers. We have extensive tables illustrating differences between segments, but there is so much data that it is difficult to get an intuitive grasp of the segments and their differences without some kind of initial orientation. It is relatively easy to become oriented by reading the narrative description that follows the tables. However, our organization of this section, placing the narrative after the tables, did not support clarity and understanding. We realized that by moving the narrative discussion before the presentation of the tables, the reader will have an intuitive understanding of the segments and have a better idea of what he or she is looking at when comparing the segments in the tables. We believe this change makes the results much easier to follow for the reader and we thank the reviewer for bringing our attention to this issue.

7. Demographic profile of one segment is different from other should have confirmed by applying ANOVA/Chi Square, t test.

The demographic profiles and other profiling variables indeed are differentiated using t-tests, as indicated with the subscripts. While we explain this in text, we omitted referring to the text explanation in a note to the table. We have corrected this omission and that should prevent future confusion on this point.

It also struck us that the reviewer is implicitly be raising a larger point, about more clearly validating the value of the segmentation approach in ways that would be more readily understood by readers from a public health background, using more conventional statistical approaches. It is an excellent point and one we are taking seriously.

Accordingly, we have added one more set of analyses, which we believe serves to further underscore the value of this contribution. In public health, and among epidemiologists in particular, there is considerable emphasis on demographic differences and demographic predictors. One would expect demographic variables such as age, income, and education in particular to be influential in predicting Covid-19 protective behaviour and vaccine refusal, given that risks were strongly age-related, income might influence a variety of risk factors such as the necessity of working in high-exposure settings, and education might be related to scientific literacy and understanding of the value of vaccination and protective behaviors. We compare a full set of demographic variables with the segmentation variable, and find that the segmentation variable is clearly the best predictor of protective behaviours and vaccination refusal (see text for the added section).

8. There is no citation included in discussion part.

In the first part of the Discussion section we discuss our findings in the light of previous findings in the literature and provide related citation. In the latter part of the Discussion we focus on managerial implications of our findings; since our discussion there is focused on our findings, we saw no utility for the reader in adding citations to that section.

9. The manuscript is not structured as per the journal guideline.

Thank you for pointing this out, we have carefully edited the ms and I believe we are now consistent with PLOS-One guidelines.

10. What is the Managerial implication of the study?

We have a lengthy section concerning implications for how to design public health communication efforts directed at these segments, on probing for additional insights, and regarding concerns regarding using such methods to segment diverse populations with different norms re responding to closed-end items. We realized on rereading that the main problem was that our headings did not clearly signal this focus and have revised a key section header in our Discussion section to read “Implications for public health managers and communicators regarding reaching these segments” and have clarified our language elsewhere in the Discussion section.

---

## [Editor Report · Decision Letter 1]

5 Dec 2023

PANDEMIC LIFEWORLDS A SEGMENTATION ANALYSIS OF PUBLIC RESPONSIVENESS TO OFFICIAL COMMUNICATION ABOUT COVID-19 IN ENGLAND

PONE-D-23-27498R1

Dear Dr. Coleman,

We’re pleased to inform you that your manuscript has been judged scientifically suitable for publication and will be formally accepted for publication once it meets all outstanding technical requirements.

Kind regards,

Prabhat Mittal, Ph.D.

Academic Editor

PLOS ONE
---

## [Editor Report · Acceptance letter]

22 Jan 2024

PONE-D-23-27498R1 

PLOS ONE

Dear Dr. Coleman, 

I'm pleased to inform you that your manuscript has been deemed suitable for publication in PLOS ONE. Congratulations! Your manuscript is now being handed over to our production team.

Kind regards, 

on behalf of

Dr. Prabhat Mittal 

Academic Editor

PLOS ONE